# Experimental Investigation of Cracking and Impact Resistance of Polymer- and Fiber-Enhanced Concrete for Ultra-Thin Whitetopping

**DOI:** 10.3390/polym14214472

**Published:** 2022-10-22

**Authors:** Liangliang Chen, Shaopeng Zheng, Xiao Li, Zhihao Cheng, Xiaonan Wang

**Affiliations:** 1Broadvision Engineering Consultants, No. 9, Shuangfeng Road, Guandu District, Kunming 650299, China; 2Harbin Institute of Technology, School of Transportation Science and Engineering, 73 Huanghe Road, Harbin 150090, China

**Keywords:** polymer-modified concrete, ultra-thin whitetopping, fiber reinforcement, cracking resistance, impact resistance test

## Abstract

In order to investigate the effectiveness of polymer modification and fiber reinforcement on the cracking and impact resistance of concrete materials prepared for ultra-thin whitetopping (UTW), carboxyl butyl benzene latex and polyformaldehyde fibers were added to the conventional cement concrete mix. In addition, test methods that used an asphalt mixture performance tester (AMPT) and mechanical rammer were developed to evaluate concrete cracking and impact resistance, respectively. Results from the AMPT test revealed that the cracking resistance can be enhanced with polymer and fiber, especially the initial tensile load peak, which can be improved by more than 40% when fiber and polymer compound modification is applied. Meanwhile, the impact loading test revealed that the inclusion of both fiber and polymer results in a two-fold increase in the number of impacts before visible cracking occurs, and the number of blows to failure increased by 21.4%. Moreover, microstructures were investigated by scanning electron microscopy (SEM) to confirm the reinforcing mechanism of both polymer modification and fiber reinforcement.

## 1. Introduction

Thin concrete overlays bonded to hot mixed asphalt (HMA) with the typical thickness limited to 5–10 cm, also known as ultra-thin whitetopping (UTW), has been used for the repair of distressed asphalt pavement in the US since its introduction in Madisonville, Kentucky, in 1991 [1]. UTW has a much thinner thickness and only 1/6–1/2 of the Jointed Plain Concrete Pavement (JPCP). In general, UTW, as the top layer of pavement but with much smaller thickness, provides higher strength and better cracking and impact resistance.

Overall, UTW can perform well and be an effective pavement rehabilitation technique for highways, parking lots, and intersections. A full bonding can be achieved when the substrate is of good quality [2,3,4]. Unfortunately, cracking problems and other deterioration mechanisms were reported by King and Roesler [5]. In view of the characteristics of low toughness, high brittleness, and poor impact resistance of Portland cement concrete (PCC), some components are added to improve the performance of overlay materials. Previous research has proved that fibers and polymer can effectively improve the toughness of PCC as well as improve durability and abrasion resistance [6]. The benefits of fiber reinforcement have been reported in many research publications (Cao et al. 2017; Mehta and Monteiro 2006). The introduction of fibers in conventional PCC can increase the tensile strength and load capacity for UTW significantly (Bordelon and Roesler 2012; Han 2005). Carboxylated styrene–butadiene latex has good compatibility with PCC materials and is commonly used as a concrete modifier [7], which can increase the setting time of PCC and improve the workability and flexural strength. Notably, polymer modification can significantly enhance the bonding strength of cement mortar and concrete for repair works—not only the bonding between the old substrate and repair concrete but also the bonding between the matrix and aggregates [8,9,10]. On the other hand, the polymer also has some negative effects, such as increasing macroporosity of concrete, thus decreasing the compressive strength [11]. Polymer modification can effectively fill the voids and defects inside hydrated cement pastes. Moreover, the polymer film can further enhance the bonding between cement hydrates aggregates and fibers. As a result, the fiber can control the cracking scale more effectively compared with fiber reinforcement only [12].

However, the properties of UTW with the addition of both fiber- and polymer-compound-modified concrete (FPMC) have been seldom researched. Therefore, further studies are needed to shed light on their properties and guide their application in civil engineering, especially for UTW.

Additionally, great efforts have been put into correlating laboratory tests results with field performance. Laboratory tests are conducted on most concerned properties of rigid pavement, such as flexural strength, modulus of elasticity, and drying shrinkage. In general, the surface layer of concrete is required to achieve the specified strength level to undertake the load from the environment and traffic. Traditionally, compressive and flexural strength are commonly used as the indicators of the life service of cement concrete. However, higher strengths lead to higher modulus of elasticity and coefficient of thermal expansion, thus increasing the curling stress of rigid pavement accordingly [13,14]. Therefore, it is pointed out by many engineers and researchers that the strength does not correlate with the pavement performance very well. Field investigation reveals the reduction in the failure level is not as dramatic as it was supposed when the flexural strength increases [15]. As the pavements are highly exposed to repeated dynamic wheel loading rather than static loading [16], the toughness and energy absorption capacity are the essential characteristics in addition to flexural strength for concrete pavement. Furthermore, insightful investigation is needed to study the resistance of cracking and impact loading. Especially notable, the thickness of UTW is much lower than conventional concrete pavement, and its fracture behavior is much more like a plate rather than a beam; thus, the fracture toughness evaluation by three-point bending tests cannot simulate the mechanical response under dynamic traffic loading in the field very well.

Compared with laboratory tests on strengths, full-scale tests are believed to evaluate the cracking resistance of different types of concrete under dynamic loading more effectively [17,18,19]. However, it is quite time-consuming and expensive.

On the other hand, varieties of impact loading tests in laboratory were conducted to evaluate cracking durability of concrete, such as the procedures proposed by ACI committee 554 (ACI 1999). According to ACI 1999, a series of drop-weight impact tests were proposed [20,21], as well as procedures that were carried out with concrete slabs [22] and other self-developed devices [23]. Unfortunately, none of these test procedures has been performed widely. Apart from experimental studies, the numerical method can be used to determine the fracture properties of composites. Among these numerical methods, the “Extended Finite Element Method” and “Bezier Multi-Step Method” can provide more reliable results [24,25].

## 2. Objectives

The objectives of this study are to further investigate the influence of fiber and polymer compound modification on the fracture toughness and impact resistance and improve the test efficiency for each of them. Firstly, polymer and fiber reinforcement were added to concrete mixes in order to improve the cracking and impact resistance. The compressive and flexural strengths were tested for conventional, polymer-modified concrete (PMC), fiber-reinforced concrete (FRC), and fiber- and polymer-compound-modified concrete (FPMC). Secondly, overlay tests by Asphalt Mixture Performance Tester (AMPT) were carried out to investigate the cracking propagation process, and the tensile peak loads were recorded. The procedure was set up as follows: (i) the controlled maximum displacement of the cyclic triangular tension loading was determined by experiment and (ii) cracking resistance and maximum tensile loads were recorded to evaluate cracking resistance.

Lastly, drop-weight impact loading tests with a mechanical rammer in accordance with ASTM D1557 (Standard Test Method for Laboratory Compaction Characteristics of Soil Using Modified Effort) were carried out as follows: (i) Using mechanical rammer, a drop-weight impact loading test procedure was developed, including the determination of the proper height of the specimen cut from a standard cube specimen for testing the compressive strength of concrete. (ii) Drop-weight loading test was carried out, and the blow number was recorded.

## 3. Materials and Methods

### 3.1. Materials

Ordinary Portland cement P∙O 42.5 produced locally in Yunnan Province, China, was used in the mix design. Its main properties can be found in Table 1. The fine aggregate was manufactured basalt sand with a fineness modulus of 2.72. The coarse aggregates were limestone crushed aggregates with a maximum size of 26.5 mm, the crushing value of which was 13.7%. High-efficiency hydroxylated carboxylic acid was added as the water reducer, the content of which was 1.2% by the cement mass.

The polymer-based modifier used in this research was carboxyl butyl benzene latex. As shown in Figure 1a, it is a milk-like emulsion and the typical properties are presented in Table 2.

Due to the surface activity of carboxylated styrene–butadiene latex, excess air bubbles will be produced in the mix, causing discontinuities within the concrete and resulting in a negative effect on the impermeability and compressive strength [26]. Therefore, a defoaming agent with good compatibility with carboxylated styrene–butadiene latex was necessary to counteract the adverse influence. In this study, foamstar 2706 (triisobutyl phosphate in 5 to 7%, CAS: 126-71-6, properties are shown in Table 3), commonly applied in cement mortar and concrete, was purchased from BASF, the adding content 3‰ by mass of cement.

In addition, polyformaldehyde fibers (see in Figure 1b) were added to both conventional and polymer-modified concrete to further improve UTW performance. The technical parameters can be found in Table 4.

### 3.2. Proportioning

All the mix designs were developed with constant water-to-cementitious-material (W/C) ratio of 0.36 and a cement amount of 420 kg/m^3^. The concrete specimens with polymer modification were prepared with the polymer/cement mass (P/C) ratio fixed to 7.5%; the determination of the P/C ratio was based on previous research in this project [27], at the level of which a maximum value of flexural strength could be achieved without significant decrease in compressive strength.

The mix proportion designs are given in Table 5, where C_0_ was the reference, without polymer modification or fiber addition; CP_1_ was modified by polymer only, while CP_2_ was polymer-modified as well as 3‰ defoaming agent. CF_1_, CF_2_ were fiber-reinforced concrete with the content of polyformaldehyde fiber 0.9 kg/m^3^ and 1.2 kg/m^3^, respectively; CPF was polymer-modified concrete with 1.2 kg/m^3^ of polyformaldehyde fiber.

To minimize the disruption to the travelling public due to work zones, pavement repair jobs are required to open to traffic as soon as possible. It is necessary that the concrete overlay materials can gain enough strength more quickly than the traditional concrete product to cut the time on curing. Aiming to open to traffic within 3~7 days, a mixture of calcium chloride and calcium nitrite (CaCl_2_:Ca(NO_2_)) = 1:1, was added as an accelerating admixture with a fixed content of 1.9% of cement mass.

The aggregates and cement were added to the mixer and mixed for 30 s, then fiber was added and mixed for another 30 s. The polymer was mixed with the mixing water firstly for the purpose of dilution and better dispersion and added into the mixer separately with the aqueous solution of the admixtures and deforming agent. Finally, another 90 s of mixing was required after adding materials.

## 4. Test Methods Development

### 4.1. Cracking Resistance Test

The overlay tester is commonly used to determine the resistance ability of asphalt mixture to reflective or fatigue cracking, and a test procedure has been introduced by Zhou [28]. In this research, a continuous fatigue damage analyzer (AMPT) was used to test the cracking resistance of the specimens after 28 days of standard curing. The test specimens were prepared according to the procedure specified by TxDOT (Texas Department of Transportation) [29]. Firstly, the standard cube test specimens for the compressive strength test were cut into samples 150 mm long, 75 mm wide, and with a height of 38 mm (Figure 2). Secondly, as shown in Figure 3, the prepared samples were glued to the test plates with two-part epoxy, the 24 h shear strength of which is 18.1 MPa. The apparatus features two separate plates, one of which is fixed, and the other one can slide vertically. An electro-hydraulic system is employed to apply repeated direct tension loads. The sliding block can apply tension in a cyclic triangular waveform to a pre-set maximum displacement, as is illustrated in Figure 4. To determine the proper maximum displacement for cementitious composite, two displacement levels were set, namely 0.1 mm and 0.2 mm, with the loading frequency of 0.1 Hz. The peak load value obtained for the first cycle was taken as the initial load. At the controlled maximum displacement of 0.1 mm and 0.2 mm, the peak load for each cycle was recorded. The test result was also interpreted by the peak load reduction rate, which worked as a relative parameter that was determined by the peak load value during the following load cycles compared with the absolute value of the first cycle at the same crack opening.

### 4.2. Drop-Weight Impact Loading Test

As presented in Figure 5, the impact loading test was carried out in accordance with ASTM Standards D 1557, with a minor revision that replaced the mold with a cramping apparatus to fix the concrete specimen during testing. The mass of the mechanical rammer is 4.5364 ± 0.009 kg, the drop height (H) from the highest point to the lowest point is 67 cm, and it can drop freely from a certain height following the height of the specimens. Suppose the thickness (T) of the specimen is known. In that case, the impact energy can be calculated by Equation (1). Therefore, the impact resistance performance of different specimens can be evaluated by the number of impacts of the concrete specimens.
(1)$$W=Nm_{r}gh_{e}$$
where *W* represents impact energy (J); *m_r_* represents mass of the mechanical rammer (kg); *g* represents acceleration due to gravity (9.81 m/s^2^); *N* represents number of blows; and *h_e_* = H − *T*, effective drop height (m).

### 4.3. Scanning Electron Microscopy (SEM)

In order to further understand the reinforcing mechanism of fiber reinforcement and polymer modification, SEM tests were conducted on the samples from groups with polymer modification only (CP), fiber reinforcement only (CF), and both fiber and polymer addition, as well as the conventional concrete. Prior to the observation, the samples should be dried and gold-plated in an ion sputter coater for 2 min. The microstructures, hydration meshes, and fiber distribution were examined with JSM-7800F scanning electron microscope; the accelerating voltage was from 2 to 15 V.

## 5. Results and Discussion

### 5.1. Strengths

The compressive strength was tested with cubic specimens (150 mm × 150 mm × 150 mm) according to ISO 4012-1978, and the load rate was set from 0.5 MPa/s to 0.3 MPa/s. The flexural strength tests were conducted according to ISO 4013-1978 or ASTM C78, and the loading rate was set from 0.05 Mpa/s to 0.08 Mpa/s. The compressive strength and flexural strength of the mixtures are shown in Figure 6a,b. The 3-day and 7-day test results indicated that the incorporation of polymer significantly reduced the compressive strength by more than 20%, and fiber-reinforced specimens in group CF_1_ and CF_2_ exhibited comparable compressive strength to the conventional concrete. Comparing test results of flexural strength of CP_1_ and CP_2_ with C_0_, it can be concluded polymer modification results in much higher flexural strength than ordinary PCC, by 36.8% on average for test results of 3-day age and by more than 38% for 7-day age. As for fiber-reinforced specimens, the increments were 6.3% and 27.5% on average for 3-day and 7-day flexural strength, respectively. By examining mixtures CP_1_ and CP_2_, it was found that adding defoaming agents can slightly improve compressive and flexural strength by approximately 7% and 4%, respectively. However, adding polyformaldehyde fiber to polymer modified concrete did not affect the improvement of compressive or flexural strength, which resulted in comparable compressive and flexural strengths for CP_2_ and CPF.

The 3-day flexural strength for all the mix designs was larger than the required value, according to Harrington and Fick [30]. The test results above showed that the flexural strength of all the mix designs could reach the desired value after curing for 3 days. Polymer modification significantly improved flexural strength compared with C_0_. However, the CP_1_ and CP_2_ with polymer modification did not achieve adequate compressive strength at the 3-day age due to the increase of macro porosity. UTW is often used as a repair material for existing asphalt concrete pavement. Therefore, its curing time is required to be as short as possible. The addition of polymer will prolong the setting time of cement concrete [26] and delay the opening to traffic of UTW. However, the negative effect can be balanced by adding accelerating admixture. 

### 5.2. Test Results of Cracking Resistance

The cracking resistance tests were carried out at maximum displacement of 0.2 and 0.1 mm. Test results for crack opening of 0.2 mm are displayed in Figure 7 and Figure 8. At the beginning of the test, it was found that the tensile load peak for maximum opening of 0.2 mm decreased drastically after the first loading cycle, which indicated that the concrete failed at tension. The tensile load peak lowered to an almost constant level after only the first five loading cycles. Therefore, the number of the whole loading sequence was reset to 60 to improve the test efficiency. At the end of loading cycles, the tensile load peak for the specimens with polymer modification did not exhibit an obvious difference compared with the conventional concrete. On the other hand, the residual strength for CF_1_, CF_2_, and CPF after cracking was attributed to the fiber reinforcement. The load reduction rate showed a similar trend. The load reduction rate of the mixture without the inclusion of fibers increased to more than 85% at the end of the first cycle and gradually reached 90% in the next few runs.

In summary, it was found that at maximum displacement of 0.2 mm, concrete failed too quickly to recognize the difference in the cracking propagation process for each mix. Therefore, the maximum displacement was reset to 0.1 mm.

Likewise, test results with a controlled maximum crack opening of 0.1 mm are displayed in Figure 9 and Figure 10. The peak load levels for C_0_ and CP_1_ were almost the same after 60 runs, then the two lines overlapped. However, polymer modification prolonged the process of breaking down, and tensile load decreased sharply to a very low level, yet it cost the CP_1_ and CP_2_ almost 20 runs to achieve a significant load reduction rate (Figure 10). Moreover, the increase of fiber content did not show a clear difference on tensile load peak before the 80th run comparing CF_1_ and CF_2_. However, the tensile load level for CF_2_ was higher than CF_1_ for the following 40 cycles. Notably, the initial tensile load for the first cycle of CPF was much higher than specimens with polymer modification only and with fiber reinforcement only.

As was pointed out by Liu (Liu et al. 2018), the rank of the peak load level of different mixtures is not necessarily consistent with that of the load reduction rate because the load reduction rate is a relative parameter that is highly dependent on the initial peak load level. Overall, the load reduction rate for the C_0_ was higher at the first few runs and kept almost constant after 20 cycles. Polymer modified ones, CP_1_, CP_2_, and CPF reached a higher load reduction rate than the conventional concrete C_0_. Moreover, concrete with only fiber reinforcement exhibited the lowest load reduction. In addition, Figure 11 and Figure 12 display plots to recognize the difference during the first 20 cycles. In Figure 12, it can be seen clearly that the tensile load of C_0_ decreased dramatically. As a result, the load reduction rate reached 22% at the end of the first loading cycle. As for polymer modification and fiber reinforcement groups, the tensile load decreased linearly for CP_1_, CP_2_, CF_1_, CF_2_, and CPF during the first 5 cycles. Moreover, compared with CF_1_ and CF_2_, it took more time for the specimens with polymer modification to reach an almost-constant loading reduction rate, which implied a better anti-cracking performance. However, the groups with only fiber reinforcement exhibited a much lower load reduction rate after the fifth loading cycle due to the toughness improvement as a result of the dispersed polyformaldehyde fibers within the concrete. A better cracking resistance at cracking occurrence could be concluded, owing to the load reduction rate after the first load cycle.

Concrete fails under tensile loading at a much lower stress level compared with compressive loading. Cracking occurs at the first loading cycle, then propagates during the following loading cycles. In fact, stress concentration at voids, in other words, defects, becomes large enough to initiate micro-cracking within the concrete. At the end of the first loading cycle, the crack will appear, and the concrete will fail in a brittle manner. Therefore, the peak load level before concrete cracking can be taken as an indicator of the micro properties of concrete.

To make an easy comparison, the conventional concrete maximum tensile load of C_0_, the reference group, was defined as 100%, and the percentage of other groups could be easily determined with respect to the reference. Initial tensile loads for the two maximum displacements are shown in Figure 13 and Figure 14. The addition of defoaming agent effectively reduced the detrimental bubbles during the mixing and hardening process of polymer modified concrete. Therefore, the defoamer significantly influenced cracking resistance in terms of making the concrete structure more homogeneous and denser. As a result, comparing the test results of CP_1_ and CP_2_, polymer modification could improve the initial tensile load peak, especially when the defoaming agent was added. In addition, fiber reinforcement improved the initial load peak by 15.6–23.2%. Additionally, suppose fiber and polymer compound modification was included. In that case, the initial tensile load peak can be improved by 45.7% on average, much higher than that with only polymer or fiber addition.

Clearly, the fiber and polymer compound modification can improve the cracking resistance significantly. The improvement of the maximum tensile load is attributed to three influence factors. Firstly, the polymer can fill the voids inside the concrete; secondly, the introduction of fiber can obstruct the propagation of the microcracks; and lastly, the carboxylated styrene–butadiene latex was an adhesion agent, which can adhere both the high-efficiency hydroxylated carboxylic acids on the surface of concrete and the polyformaldehyde fiber.

### 5.3. Impact Loading Test Results

Impact resistance tests were performed after the concrete was cured for 28 days at 20 °C, 95% RH. In this study, the standard cube specimens of 150 × 150 × 150 mm were tested directly without trimming. Unfortunately, all the specimens for the seven groups sustained more than 500 impacts prior to cracking occurrence. Moreover, there was no cracking found for C_0_ even with number of blows up to 5000 times. As it is shown in Figure 15a,b, the rammer penetrated into the specimen after 5000 blows of impact loading, but no cracking occurred. In other words, trial test results showed that the impact loading with 150 × 150 × 150 mm specimens was too time-consuming to evaluate the impact resistance of different mixtures. Therefore, to improve the test efficiency of impact loading, the standard cube specimens were trimmed with a sawing device to different sizes of 150 × 150 × *T* mm, where *T* was the thickness of trimmed specimens under the rammer, which varied at 150 mm, 120 mm, 100 mm, 70 mm, 50 mm, and 35 mm.

Impact resistance test results for different thicknesses of C_0_ are shown in Figure 16. For the specimens with thickness less than 100 mm, the number of blows to failure was less than 10, and no convincing difference on the number of impacts was found among the specimens with thickness of 35 mm, 50 mm, and 70 mm; thus, thickness less than 100 mm was not suitable to be chosen as the thickness after trimming. In contrast, the impact loading times to failure increased drastically when the thickness of the specimen was larger than 100 mm. As a result, the average number of blows for the specimen with the thickness of 100 and 120 mm were 84 and 260, respectively. Both of them were acceptable to be the trimming standard.

Number of blows is assumed to be a power function of thickness of the specimens for the specimens that has distressed, namely thickness ranging from 35 mm to 120 mm. Then, the following regression equation (Equation (2)) can be used to describe the relationship between the thickness and the number of impacts to failure.
(2)$$N=3.046\times 10^{\left(-11\right)}\times T^{6.219},R^{2}=0.9991$$
where *N* is the impact loading times to failure and *T* is the thickness of the concrete specimens (mm).

It can be seen that the impact loading times are a function of the 6.219th power of the thickness, which means a small amount increase in the thickness can increase the number of impacts to failure greatly. For instance, if the thickness is increased from 130 mm to 140 mm, then the impact times inducing it to failure will increase from 426 to 676, making the test much more time-consuming.

As a result, 100 mm was adopted as the optimum thickness for trimming specimens due to its higher efficiency and lower variance level than 120 mm. The standard specimen with a thickness of 150 mm, and the trimming specimen with a thickness of 100 mm is displayed in Figure 17.

Impact loading tests were performed on concrete specimens of different mix proportions but of the same size (150 × 150 × 100 mm). During the test process, cracking behavior was monitored, and the test paused immediately when visible cracking was detected; the crack width was then measured and recorded. The impact loading restarted and stopped until the failure of the specimen. One of the major failure modes was corner cracking, a fissure extending from one side to the adjacent one, as shown in Figure 18a, and sometimes spalling on the surface as well. The other failure mode was trans-cracking, a crack that went through from one side to the opposite side, as shown in Figure 18b. Interestingly, several specimens of C_0_ broke into pieces, shown in Figure 18c.

Once the crack initialed and could be observed, the drop-weight test paused, and the measurement of the cracks was carried out along the crack for four positions; the maximum reading was then recorded as the crack width for the specimen. The average values for the four specimens in each group are listed in Table 6. Maximum initial cracking width was found in group C_0_, reaching 0.9 mm. Corner cracks appeared in group CP_1_ which represented specimens with polymer only, and the average crack width reduced to 0.2 mm; average crack width of CP_2_ specimens with polymer modification and defoamer addition further reduces to 0.15 mm, indicating that the use of defoamer could increase the compactness of concrete, which was beneficial to the improvement of impact resistance of concrete. Fiber reinforcement was capable of reducing the crack width. The average crack width was reduced from 0.35 mm to 0.20 mm when the fiber content was increased from 0.9 kg/m^3^ to 1.2 kg/m^3^. The average cracking width of CPF was notably reduced, such as to 0.10 mm. The outstanding effect on the improvement of fatigue performance could be predicted based on crack width measurement.

The number of blows was recognized in two series. One was the number of blows when the first crack can be observed, indicating the initiation of cracking. The other was recorded when the specimen was broken into two or more pieces, indicating the failure of the specimen.

As shown in Figure 19, by comparing group C_0_ with others, results indicated that the inclusion of polymer alone could significantly improve the impact resistance of cement concrete. In addition, fiber reinforcement could effectively delay the initiation of crack and prolong the life to failure, whereas it did not exhibit any improvement on the impact times to failure. Comparing CPF and CP_1_, the impact resistance improved most with the combination of polymer and fiber, which doubled the number of impacts to induce the first visible cracking and improved the number of impacts to failure by 21.4%. Although the number of impacts for initial crack was almost the same for CP_1_ and CP_2_, the addition of the defoaming agent did improve the life to failure by 27.8% compared with the specimens without it. Polymer latex significantly improved the interfacial bond between matrix and fibers, thus making the fibers more difficult to slide out during the impact loading and acting as crack-bridging, which was beneficial to the energy dissipation under impact loading conditions. The cracking resistance of cement concrete could be improved by simply adding fibers to conventional concrete. Due to the bridging effect of fibers in the previous section, the number of impacts until cracking came forth was significantly improved comparing CF_1_ and CF_2_ with C_0_. However, due to the small amount of fiber content and its adverse effect on the workability of fresh concrete, the impact resistance of concrete decreased.

ASTM standards are widely adopted by many countries to determine optimum water content and maximum dry unit weight for compacting soil and gravels, and the equipment is easy to find. Thus, the testing procedures developed in this research have the potential to be a standard drop-weight impact loading test in the future since it is convenient and easy to obtain the test equipment.

### 5.4. SEM Images

Figure 20 shows the SEM images of different preparation measures of cement concrete. It can be seen that there are many micro cracks and defects inside the conventional concrete (Figure 20a). With polymer modification, the phenomenon of micro cracking was reduced significantly, but only a few pores during the mixing are observed for concrete with polymer modification (Figure 20b), and it can be concluded that the polymer can fill the defects; Figure 20c shows that when only fiber reinforcement was used, there is debonding between the fiber and hydration meshes detected as well as micro cracks that can be seen clearly. With the application of both polymer and fiber (Figure 20d), the polymer can remove the defects within conventional concrete. Furthermore, the fiber penetrates into the hydration meshes much more firmly than that without polymer modification.

## 6. Discussion

Test results above indicated that the polymer modification had advantages in improving the mechanical properties of UTW materials; this agrees with the findings of the literature [11,31] that the presence of polymer can improve the bonding between fibers, aggregates, and cement greatly. In addition, when polymer was put into worksites simultaneously with fibers and defoamer, all the advantages became more pronounced in spite of the reduction in compressive strength. The findings of this research proved lower compressive strength but better fracture toughness and impact resistance. However, as the tests were carried out only in the laboratory, full-scale tests need to be performed in subsequent research to better predict the performance of UTW in the field.

## 7. Conclusions

The influence of fibers and polymer on concrete cracking and impact resistance was investigated in this research. This research presents new test procedures for cracking and impact loading resistance. On the basis of the test results above, the following conclusions can be drawn:In spite of the adverse effect on compressive strength, carboxylated styrene–butadiene exhibited a larger improvement in flexural strength compared with polyformaldehyde fibers.AMPT overlay test was proven to be capable of evaluating the cracking resistance of concrete material. Since a 0.2 mm maximum crack opening caused a sharp drop of tensile peak within only a few runs, 0.1 mm was suggested as the controlled maximum displacement.Mechanical rammer, according to ASTM Standards D 1557, was effective in performing the impact loading test with only a minor revision, and the standard cube specimens needed to be trimmed from 150 mm to 100 mm thick to improve testing efficiency.Fiber reinforcement was beneficial to improving flexural strength and inhibiting the formulation of cracking formulation but did not exhibit any advantage on the improvement of the number of impacts to failure.Fibers and polymer compound modification could significantly improve flexural and tensile strength, both of which were increased by more than 40%. Notably, the impact times of FPMC for cracking formulation was eight times that of the conventional concrete, indicating a longer fatigue life compared with conventional concrete.The defoaming agent was essential to polymer-modified concrete, which had a positive influence on the mechanical properties—not only on the compressive and flexural strength but also on the cracking resistance and impact resistance.SEM results show that the polymer modification can fill the defects inside the concrete and make the fiber stick into hydration meshes much more firmly. Furthermore, it explains the mechanical properties improvement within this research.

## Figures and Tables

**Figure 1 polymers-14-04472-f001:**
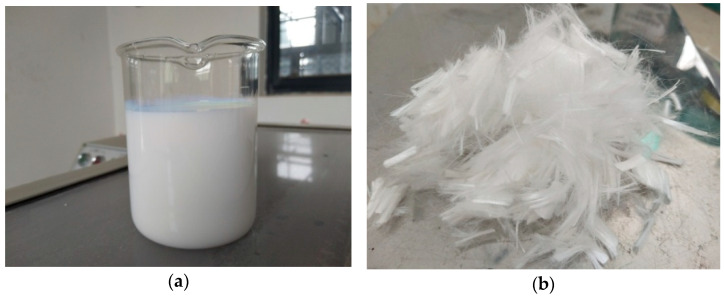
Appearance of Polymer and Fiber. (**a**) Carboxylated styrene–butadiene latex; (**b**) Polyformaldehyde fibers.

**Figure 2 polymers-14-04472-f002:**
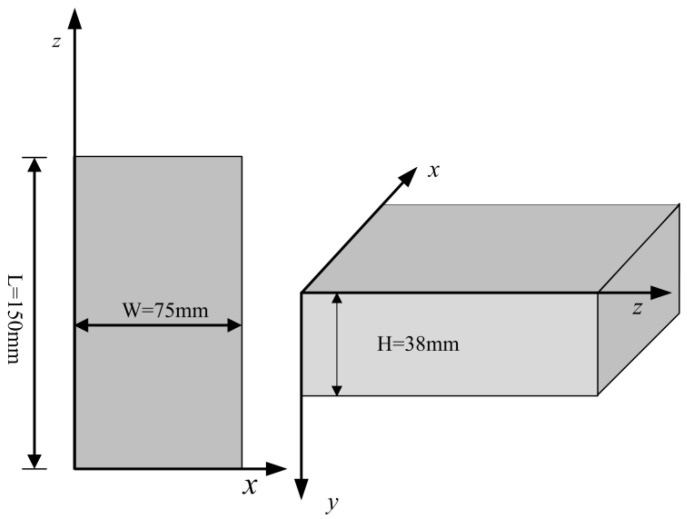
Cutting Template.

**Figure 3 polymers-14-04472-f003:**
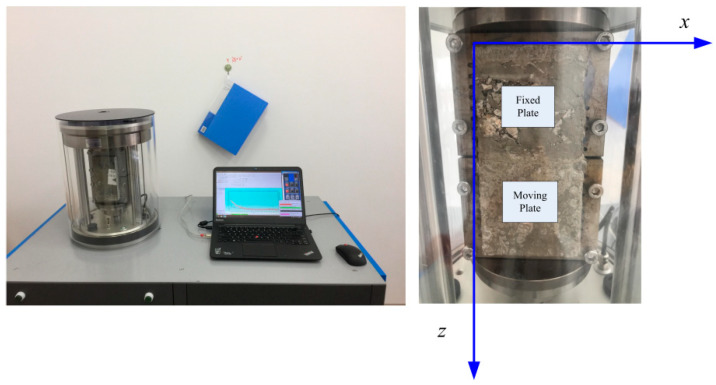
Installment of specimen.

**Figure 4 polymers-14-04472-f004:**
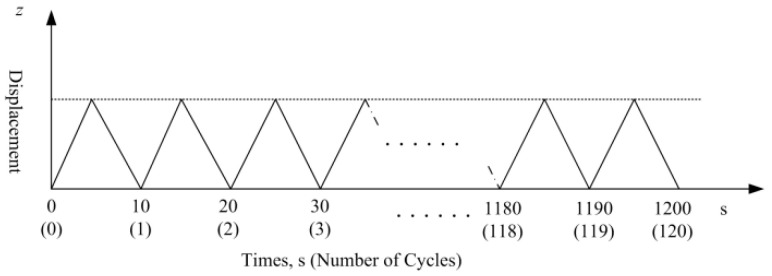
Schematic of test sequence. Note: At a maximum crack opening of 0.2 mm, the total number of cycles is 60.

**Figure 5 polymers-14-04472-f005:**
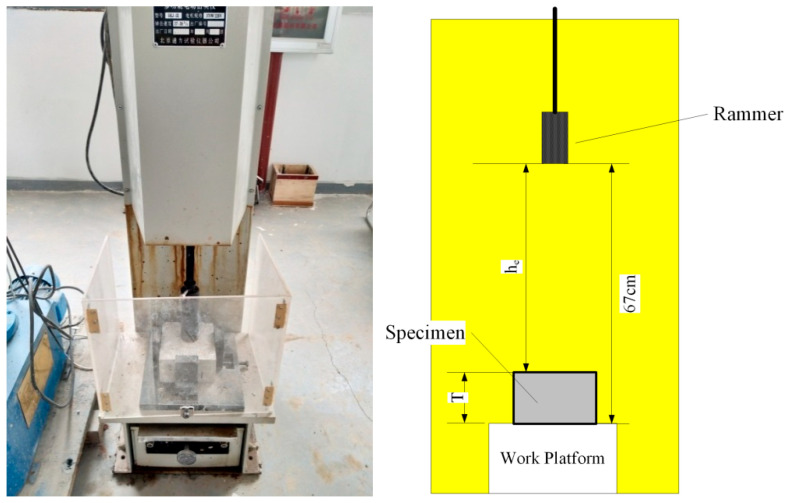
Setup of the Drop-Weight Impact Test.

**Figure 6 polymers-14-04472-f006:**
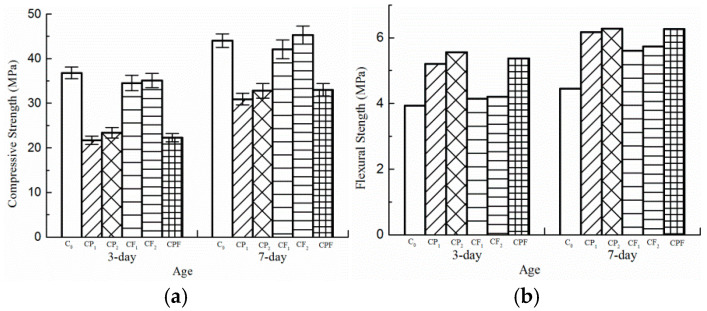
Strength test results. (**a**) Compressive strength; (**b**) Flexural strength.

**Figure 7 polymers-14-04472-f007:**
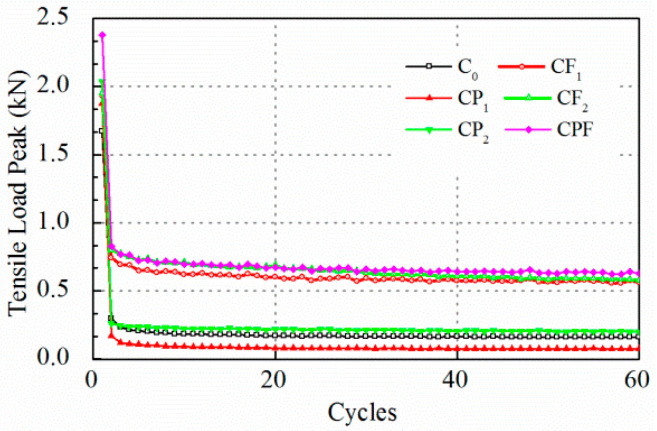
Tensile load peak at 0.2 mm crack opening.

**Figure 8 polymers-14-04472-f008:**
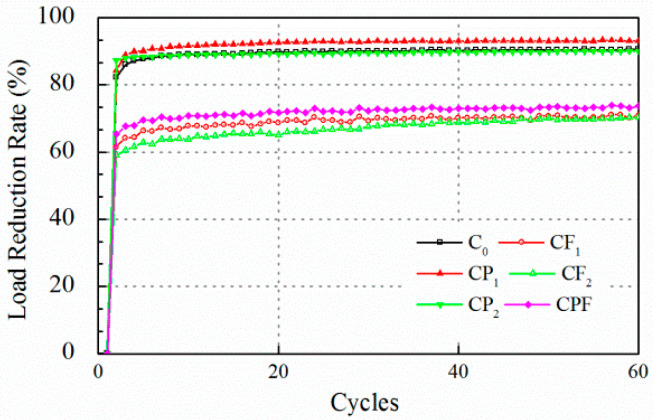
Load reduction rate at 0.2 mm crack opening.

**Figure 9 polymers-14-04472-f009:**
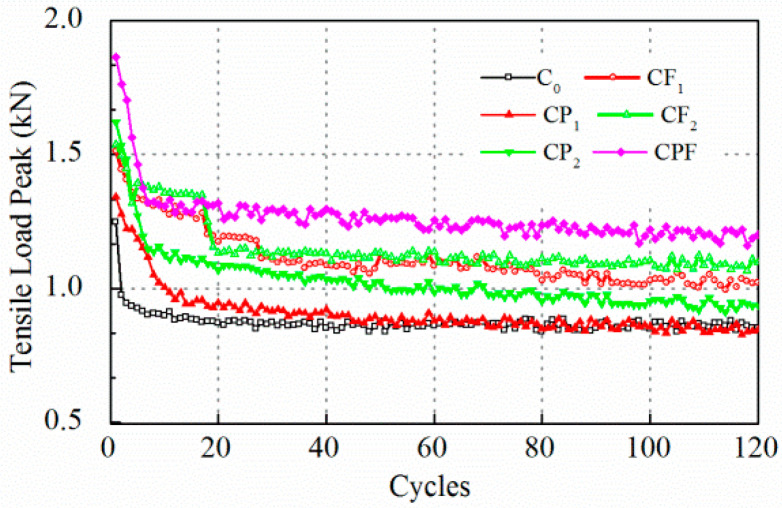
Tensile load peak at 0.1 mm crack opening.

**Figure 10 polymers-14-04472-f010:**
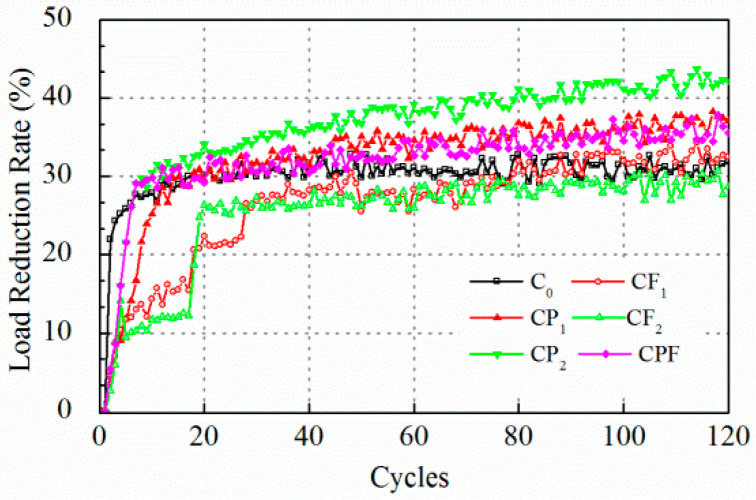
Load reduction rate at 0.1 mm crack opening.

**Figure 11 polymers-14-04472-f011:**
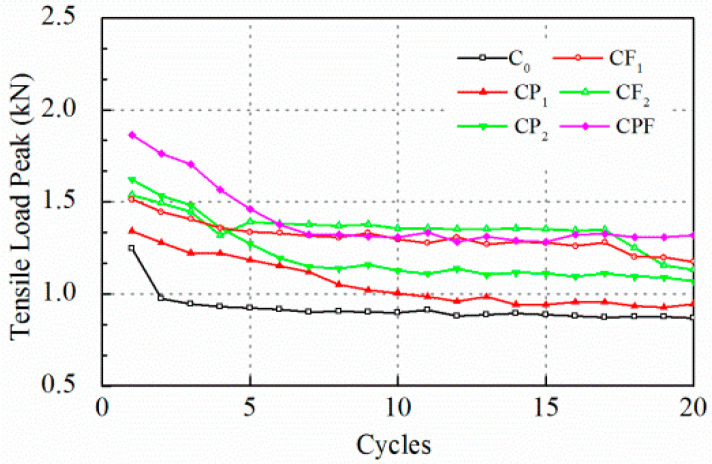
Tensile load peak at 0.1 mm crack opening for the first 20 runs.

**Figure 12 polymers-14-04472-f012:**
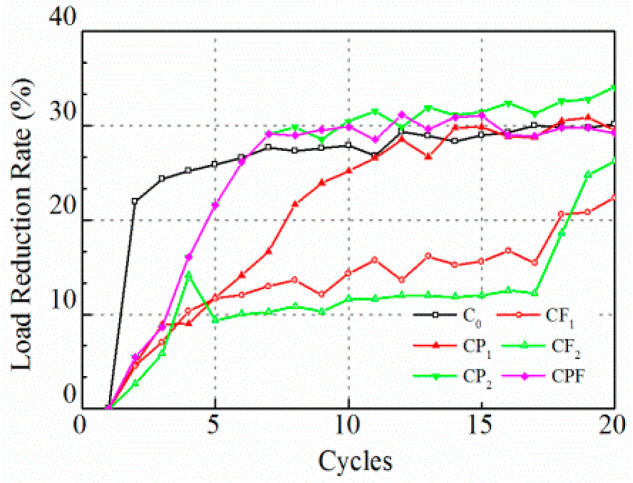
Load reduction rate at 0.1 mm crack opening for the first 20 runs.

**Figure 13 polymers-14-04472-f013:**
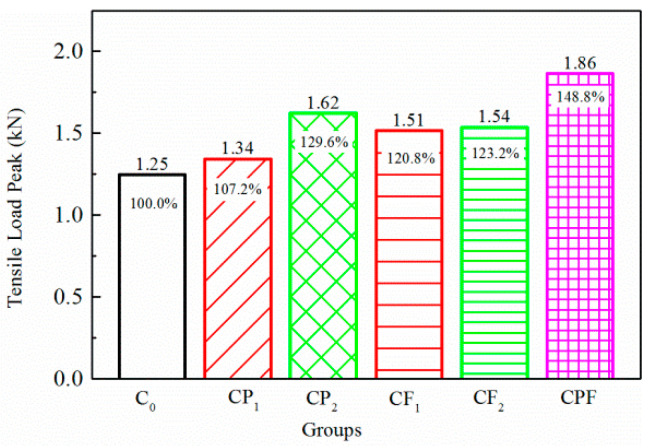
Initial tensile load for 0.1 mm crack opening.

**Figure 14 polymers-14-04472-f014:**
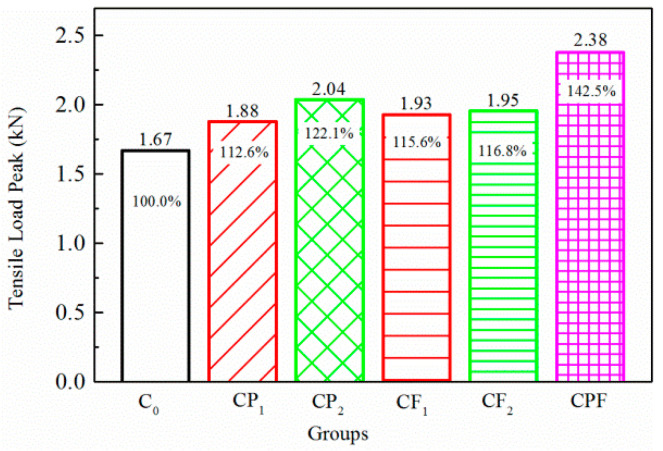
Initial tensile load for 0.2 mm crack opening.

**Figure 15 polymers-14-04472-f015:**
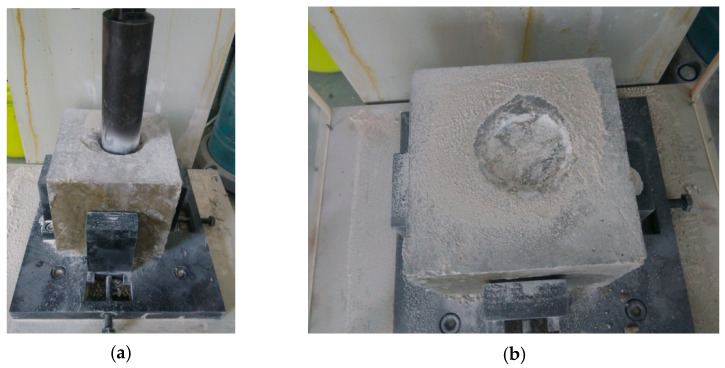
Appearance of specimens after 5000 blows. (**a**) Rammer penetrates into concrete; (**b**) State of specimen after 5000 blows.

**Figure 16 polymers-14-04472-f016:**
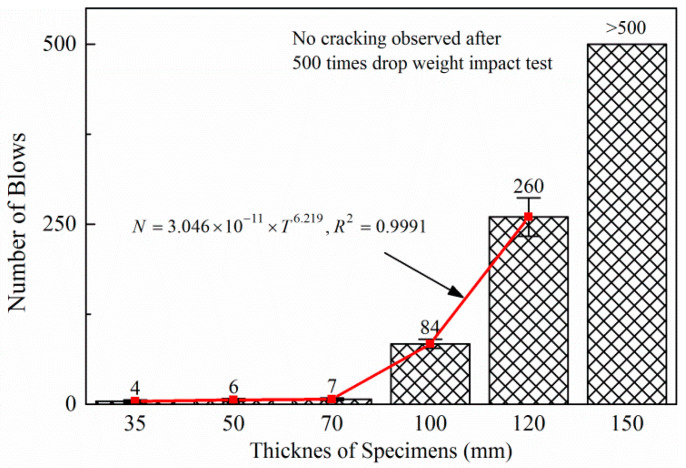
Impact loading test result for different thicknesses.

**Figure 17 polymers-14-04472-f017:**
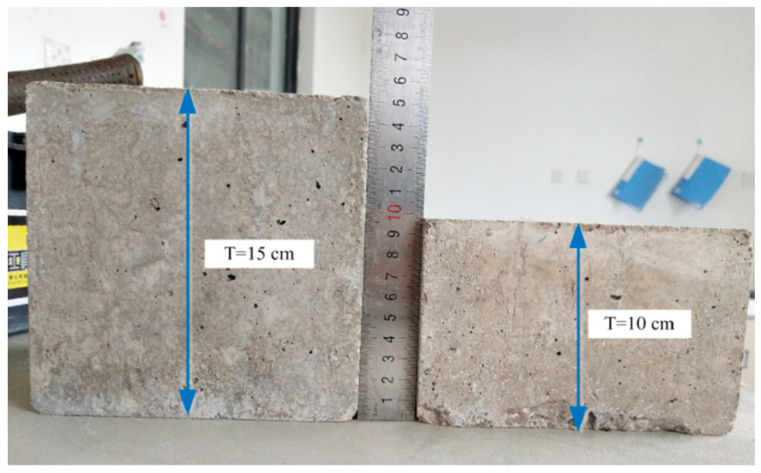
Standard and trimming specimens.

**Figure 18 polymers-14-04472-f018:**
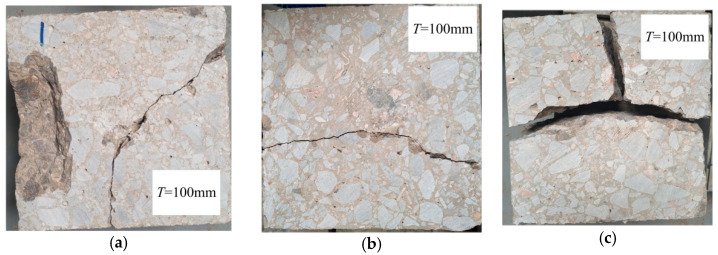
Failure modes. (**a**) Corner cracking with spalling; (**b**) Trans-cracking; (**c**) Fractured specimen.

**Figure 19 polymers-14-04472-f019:**
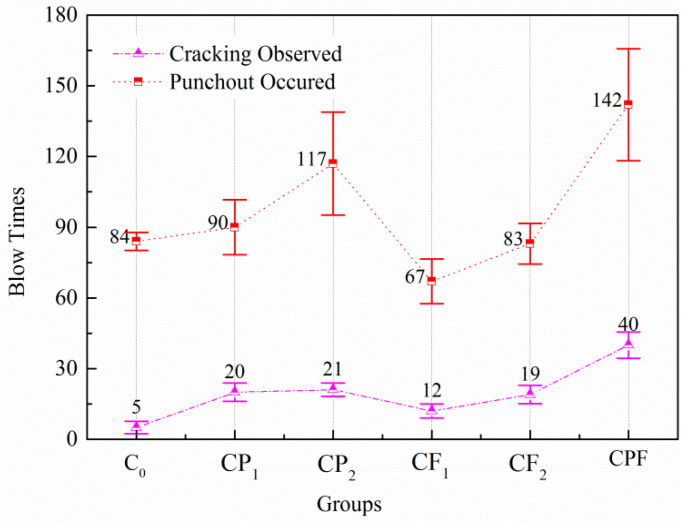
Impact test results of the same size (150 × 150 × 100 mm).

**Figure 20 polymers-14-04472-f020:**
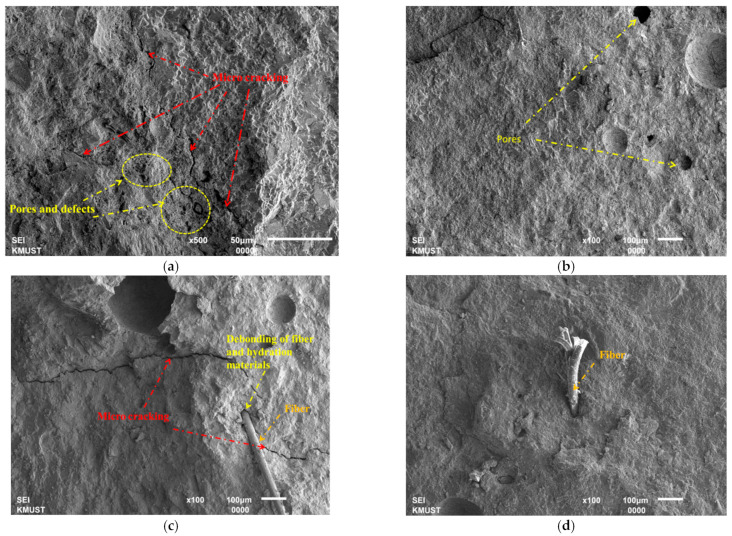
(**a**) SEM image of conventional concrete. (**b**) SEM image of polymer modification only. (**c**) SEM image of fiber reinforcement only. (**d**) SEM image of fiber reinforcement and polymer modification.

**Table 1 polymers-14-04472-t001:** Physical and mechanical properties of cement.

Fineness (%)	Specific Surface Area (m^2^/kg)	Standard Consistency Water Demand (%)	Setting Time (min)		Flexural Strength (MPa)		Compressive Strength (MPa)	
			Initial Setting Time	Final Setting Time	3d	28d	3d	28d
21	347	24	129	203	4.9	8.7	29.8	49.5

**Table 2 polymers-14-04472-t002:** Properties of carboxyl butyl benzene latex.

Appearance	Total Solid Mass Fraction (%)	PH Value	Viscosity (25 °C) (mPa.s)	Residual Mass Fraction of Styrene (%)
Milky white viscous liquid with slight fluorescence	49.25	8.25	190	0.0001

**Table 3 polymers-14-04472-t003:** Technical parameters of defoaming agent.

Characteristics	Value (Description)
Storage temperature, °C	10–30
pH value (20 g/L, 20 °C)	4
Flash point, °C	>100
Density, g/cm^3^ (20 °C)	0.99
Water solubility	Emulsified
Chemical name	Triisobutyl phosphate

**Table 4 polymers-14-04472-t004:** Technical parameters of polyformaldehyde fibers.

Characteristics	Test Results
Diameter (mm)	0.036
Length (mm)	18
Specific gravity	1.40
Fracture strength (cN/dtex)	5.2
Fiber number, dtex	14
Modulus of elasticity, (MPa)	8800
Elongation, (%)	11–14
Coefficient of variation (diameter), %	±3

**Table 5 polymers-14-04472-t005:** Mix proportion for tests.

Mixture Number	Mixture Designation	Cement (kg/m^3^)	Water (kg/m^3^)	Fine Aggregate (kg/m^3^)	Coarse Aggregate (kg/m^3^)	Water Reducer (kg/m^3^)	Accelerating Admixture (kg/m^3^)	Polymer Emulsion (kg/m^3^)	Defoaming Agent (kg/m^3^)	Fiber (kg/m^3^)	W/C Ratio
C_0_	P0D0F0	420	119.23	638.6	1185	5.04	7.98	0	0	0	0.36
CP_1_	P15D0F0	420	119.23	638.6	1185	5.04	7.98	63	0	0	0.36
CP_2_	P15D7F0	420	119.23	638.6	1185	5.04	7.98	63	1.26	0	0.36
CF_1_	P0D0F0.9	420	119.23	638.6	1185	5.04	7.98	0	0	0.90	0.36
CF_2_	P0D0F1.2	420	119.23	638.6	1185	5.04	7.98	0	0	1.20	0.36
CPF	P15D7F1.2	420	119.23	638.6	1185	5.04	7.98	63	1.26	1.20	0.36

**Table 6 polymers-14-04472-t006:** Initial crack widths for different mixtures.

Mixture Number	Average Crack Width (mm)	Variance
C_0_	0.90	0.0062
CP_1_	0.22	0.0030
CP_2_	0.15	0.0016
CF_1_	0.35	0.0032
CF_2_	0.20	0.0020
CPF	0.10	0.0012

## Data Availability

Some or all data, models, or code generated or used during the study are available from the corresponding author by reasonable request.

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
