# Peer review of "Experimental Investigation of Cracking and Impact Resistance of Polymer- and Fiber-Enhanced Concrete for Ultra-Thin Whitetopping"

_polymers, 2022, doi:10.3390/polym14214472_

Round 1

Reviewer 1 Report

The paper can be accepted after minor modification

Author Response

Please see the word "Response to Reviewer 1"

Reviewer 2 Report

Authors carried out research to investigate the effectiveness of polymer modification and fiber reinforcement on the cracking and impact resistance of concrete materials prepared for Ultra-thin white topping (UTW), which is designed to undergo high average daily truck traffic over the deteriorated asphalt pavement, whereas the thickness of UTW is reduced to only 5~10cm. Test methods that used asphalt mixture performance tester (AMPT) and mechanical rammer to were developed to evaluate the cracking and impact resistance of concrete respectively. Meanwhile, scanning electron microscopy (SEM) was used to investigate the reinforcing mechanism of both polymer modification and fiber reinforcement. Authors are claiming the addition of both carboxyl butyl benzene latex and polyformaldehyde, the mechanical performance such as flexural strength, cracking, and impact resistance was significantly improved. The study is thorough and well designed. However, there are a few concerns that should be addressed by the authors.

  1. Table 2 (line 1) its pH value, not PH value.
  2. Table 3 (line 3) its pH value, not PH value.
  3. Standard deviation values are missing for compressive Strength and flexural Strength (figure 2).
  4. Year has not been provided in reference number 17.
  5. Reference 21 is not as per journal guidelines.
  6. Discussion section 424-433 has been written poorly without coherence.  
  7. This manuscript is scientifically sound. However, the level of written English is poor.
  8. In the results and discussion section, nowhere have the authors compared their scientific findings to any reported literature.  
  9. Authors have discussed only the findings in the manuscript. They should try to explain the phenomena with evidence and literature references.  

Reviewer 3 Report

Please read and “fully” address the comments listed below:

  1. The ABSTRACT is not written in a logical order. Start with an overview of the topic and a rationale for your paper. Describe the methodology you used and the general outline of the manuscript. Also, in the end, state the result in more detail (i.e., provide some numbers).

  2. The novelty of your work is still unclear to reader, which should be detailed both in the Abstract and Introduction. In Section 3.1, please run both the XRF (find elemental composition of TPU powder) and XRD (find phase identification of a crystalline material) tests on the cement powder, and summamerize the results in a table.

  3. Please fully introduce the elastic properties of all the structural components explained in section 3.1. You can summarize them in a table.

  4. Include XYZ coordinates to all figure, e.g., in Figs. 2, 3, 4, 15, and 17.

  5. Please add error bar to the initial tensile load results shown in Figs. 9 and 10.

  6. Can you provide more close-up images of the failure modes shown in Fig. 18 and write few sentences for each mode (i.e., shear, tension, splitting, etc.)?

  7. Please provide more explanation for this sentence: Page 8, Line 235: "thus delay the opening to traffic of UTW, however the negative effect can be balanced by adding accelerating admixture."

  8. In this study, the cracking and impact resistance of concrete materials prepared for Ultra-thin whitetopping has been "experimentally" determined, which is valuable. However, apart from experimental analysis, there are strong "numerical" methods, which can be used to determine the fracture properties of composites. Among these numerical methods, "Extended Finite Element Method" and "Bezier Multi-Step Method" provide higher stability than other numerical method. Therefore, read the following selected papers, and introduce/ reference them in your manuscript, which can be used to "alternatively" determine the cracking and impact resistance of concrete materials:

    "Extended Finite Element Method": 

  • Gao, Y., Liu, Z., Wang, T., Zeng, Q., Li, X., & Zhuang, Z. (2019). XFEM modeling for curved fracture in the anisotropic fracture toughness medium. Computational Mechanics63(5), 869-883.

    "Bezier Method:

  • Kabir, H., & Aghdam, M. M. (2021). A generalized 2D Bézier-based solution for stress analysis of notched epoxy resin plates reinforced with graphene nanoplatelets. Thin-Walled Structures, 169, 108484. 

9. Conclusion: Can authors highlight future research directions and recommendations? Also, highlight the assumptions and limitations (e.g 1-2 shortcoming(s) of the present study). Besides, recheck your manuscript and polish it for grammatical mistakes (you can use “Grammarly” or similar software to quickly edit your document).

Round 2

Reviewer 3 Report

The authors addressed my comments and the manuscript can be published in the present format.